# Long-term adaptation following influenza A virus host shifts results in increased within-host viral fitness due to higher replication rates, broader dissemination within the respiratory epithelium and reduced tissue damage

Julien A. R. Amat[1,2], Veronica Patton[2], Caroline Chauché[1,3], Daniel Goldfarb[1], Joanna Crispell[1], Quan Gu[1], Alice M. Coburn[1], Gaelle Gonzalez[1,4], Daniel Mair[1], Lily Tong[1], Luis Martinez-Sobrido[5], John F. Marshall[2], Francesco Marchesi[2], Pablo R. Murcia[1]*

1 MRC-University of Glasgow Centre for Virus Research, Institute of Infection, Immunity and Inflammation, College of Medical, Veterinary and Life Sciences, University of Glasgow, Glasgow, Scotland, United Kingdom, 2 School of Veterinary Medicine, College of Medical, Veterinary and Life Sciences, University of Glasgow, Glasgow, Scotland, United Kingdom, 3 Centre for Inflammation Research, University of Edinburgh, The Queen's Medical Research Institute, Edinburgh, Scotland, United Kingdom, 4 Ecole Nationale Vétérinaire d'Alfort, Université Paris-Est, Maisons-Alfort, France, 5 Texas Biomedical Research Institute, San Antonio, Texas, United States of America

* Pablo.Murcia@Glasgow.ac.uk

**Data Availability Statement:** All the transcriptomics data is publicly available at NCBI

## Abstract

The mechanisms and consequences of genome evolution on viral fitness following host shifts are poorly understood. In addition, viral fitness -the ability of an organism to reproduce and survive- is multifactorial and thus difficult to quantify. Influenza A viruses (IAVs) circulate broadly among wild birds and have jumped into and become endemic in multiple mammalian hosts, including humans, pigs, dogs, seals, and horses. H3N8 equine influenza virus (EIV) is an endemic virus of horses that originated in birds and has been circulating uninterruptedly in equine populations since the early 1960s. Here, we used EIV to quantify changes in infection phenotype associated to viral fitness due to genome-wide changes acquired during long-term adaptation. We performed experimental infections of two mammalian cell lines and equine tracheal explants using the earliest H3N8 EIV isolated (A/equine/Uruguay/63 [EIV/63]), and A/equine/Ohio/2003 (EIV/2003), a monophyletic descendant of EIV/63 isolated 40 years after the emergence of H3N8 EIV. We show that EIV/2003 exhibits increased resistance to interferon, enhanced viral replication, and a more efficient cell-to-cell spread in cells and tissues. Transcriptomics analyses revealed virus-specific responses to each virus, mainly affecting host immunity and inflammation. Image analyses of infected equine respiratory explants showed that despite replicating at higher levels and spreading over larger areas of the respiratory epithelium, EIV/2003 induced milder lesions compared to EIV/63, suggesting that adaptation led to reduced tissue pathogenicity. Our results reveal

(accession number PRJEB29313): https://www.ncbi.nlm.nih.gov/bioproject/PRJEB29313.

**Funding:** PRM was supported by the Medical Research Council of the United Kingdom (Grant MC_UU_12014/9), the Horserace Betting Levy Board (Grants 779 and 797), the Biotechnology and Biological Sciences Research Council (Grants BB/V002821/1 and BB/V004697/1). JARA was supported by the Horserace Betting Levy Board and the University of Glasgow School of Veterinary Medicine VetFund, the Georgina Gardner Endowment (grant number 145813-01) and the John Crawford endowment (grant number 123939-01). QG was supported by the Medical Research Council of the United Kingdom (grant MC_UU_12014/12). The funders had no role in study design, data collection and analysis, decision to publish, or preparation of the manuscript.

**Competing interests:** The authors have declared that no competing interests exist.

previously unknown links between virus genotype and the host response to infection, providing new insights on the relationship between virus evolution and fitness.

## Author summary

As viruses are obligate intracellular pathogens, their ability to replicate and spread within their hosts is key for survival, even if it leads to severe disease or death of the host. Understanding the consequences of long-term virus adaptation after viral emergence is key for pandemic preparedness. H3N8 equine influenza virus (EIV) originated in birds and has circulated in horses since 1963, thus providing unique opportunities to study virus adaptation. We compared the replication kinetics of two EIVs of the same lineage but with different evolutionary histories: the earliest virus (EIV/63, isolated in 1963), and EIV/2003, which was isolated after 40 years of continuous circulation in horses. Experimental infections of cell lines (MDCK and E.Derm cells) and equine respiratory explants show that EIV evolved towards enhanced replication and cell-to-cell spread; but reduced tissue damage, confirming that viral fitness is adaptive and does not necessarily result in higher virulence.

## Introduction

Cross-species viral infections are caused by viruses of diverse families and affect a broad number of species. They range from spillover events (i.e. infections with no onward transmission) as illustrated by rabies virus (*Rhabdoviridae*) infections in humans [1], to the establishment of new endemic viruses as observed with severe acute respiratory syndrome 2 virus (*Coronaviridae*) [2]. Such varied epidemiological consequences of cross-species transmissions also apply to influenza A viruses (IAVs, *Orthomyxoviridae*). The main natural reservoir of IAVs is found in wild birds [3] and transmission of avian-origin IAVs to mammals generally results in dead-end infections (e.g. H5N1 or H7N9 IAV infections in humans [4]). However, new endemic IAVs of avian origin have emerged sporadically in mammals, as observed with H3N2 canine influenza virus [5,6]. While there are over 5000 extant mammalian species [7], the number of mammalian hosts that support the circulation of IAVs is remarkably low and includes -but is not limited to- humans, pigs, dogs and horses.

Viral fitness -defined as the ability of a virus to generate infectious progeny in a given environment [8]- is difficult to measure in a comprehensive manner. For this reason, most studies aiming at comparing fitness between viruses often measure either individual or multiple *fitness components*. For example, virus replication can be easily measured, and when applied to *in vitro*, *ex vivo* and/or *in vivo* systems, provides an accurate estimation of replicative fitness in well-defined and highly controlled environments, such as cells, tissues, or individuals, respectively. Approaches involving the quantification of replicative fitness led to fundamental insights on the relationship between virus evolution, fitness and adaptation to potentially adverse environments such as new hosts [9] antiviral drugs [10] or neutralizing antibodies [11].

As viruses are obligate intracellular pathogens, completion of their cellular replication cycle relies on molecular interactions with host proteins, which are required not only for the replication process *per se*, but also to overcome the host antiviral defences (including intrinsic, innate and adaptive immunity) and, in some cases, compete for resources with other pathogens that exhibit the same tropism [12]. A comprehensive understanding of how viruses evolve to maximise their fitness requires the combination of experimental and evolutionary approaches linking pathogen genotype to infection phenotype. The latter is significantly influenced by within

host-fitness and includes virus replication kinetics, virus spread within tissues and tissue pathogenicity. While advances in sequencing technologies and phylogenetics have generated a vast body of knowledge on the genomic evolution of viruses and its impact on viral emergence [13], it is not entirely clear how evolution impacts on virus-host interactions and contributes to the establishment of new endemic viruses.

Interspecies transmission of avian-origin IAVs into horses occurred on various occasions [14–17]. H3N8 equine influenza virus (EIV) is the only subtype currently circulating amongst equine populations, as other avian-origin EIVs such as the H7N7 subtype became extinct. H3N8 EIV was first reported in USA in 1963 [15] during an outbreak of highly transmissible respiratory disease that led to a panzootic and the establishment of EIV as an endemic lineage in several countries [18–21]. Equine influenza is characterised by lesions predominantly in the upper and middle respiratory tract, such as rhinitis and tracheitis, where virus antigen is often detected [22]. Bronchopneumonia -inflammation of lower airways and pulmonary parenchyma- has also been associated with equine influenza albeit is thought to be caused by secondary bacterial infection [22]. H3N8 EIV binds preferentially to α2,3 N-glycolylneuraminic acid (NeuGcα2,3Gal), which is highly abundant in the tracheal epithelium of the horse [23].

H3N8 EIV constitutes an ideal model to study the long-term adaptation of an avian-origin IAV that established a pandemic in a mammalian host as it has been circulating uninterruptedly in horses for almost 60 years and phylogenetic analyses indicate that this monophyletic lineage originated in birds [24–26]. Studies based on the analyses of viral genomes showed that early and contemporary H3N8 EIVs are separated by significant phylogenetic distance. This is illustrated by the large number of amino acid changes observed in the main trunk of the phylogenies of each viral genomic segment [27]. Further, some of those mutations showed evidence of positive selection as well as adaptive evolution [18,27], consistent with the notion that virus adaptation is polygenic. Experimental approaches identified the effect of individual mutations on H3N8 EIV antigenicity: mutations in the haemagglutinin (HA) gene led to immune escape from vaccine strains via antigenic drift [28], which in turn increased the probability of EIV transmission [29]. In addition, we showed that two amino acid changes that became fixed in the non-structural protein 1 (NS1) over a 20-year period had a significant impact on innate immune evasion, enabling EIV to block the induction of interferon-stimulated genes [30]. However, those studies did not capture the combined impact of multiple mutations across different genomic segments on viral fitness, which could result by the additive effect of individual mutations or enhanced by epistatic effects.

The main objective of this study was to assess the impact of long-term adaptation on within-host virus fitness. We hypothesized that the accumulation of mutations across the entire H3N8 EIV genome, as a consequence of continuous evolution in horses, resulted in changes in virus-host interactions that would impact virus-induced pathology and overall fitness. To test this hypothesis, we combined classical virology, transcriptomics, and image analysis to quantify and compare the *in vitro* and *ex vivo* infection phenotypes of two EIVs (A/equine/Uruguay/63 [EIV/63] and A/equine/Ohio/2003 [EIV/2003]). The selected viruses represent the earliest isolated virus and a contemporary virus of this monophyletic avian-origin lineage, separated by 40 years of continuous evolution in a mammalian host.

## Results

### EIV/63 and EIV/2003 exhibit significant differences in replication kinetics, cell-to-cell spread and susceptibility to interferon *in vitro*

To identify differences in virus replication between two evolutionary distinct EIVs *in vitro* that could be linked to virus adaptation, we used EIV/63 and EIV/2003 to infect two mammalian

cell lines: Madin-Darby Canine Kidney (MDCK) cells and equine dermal fibroblasts (E.Derm) cells. The former is a cell line of canine origin that is permissive to a broad range of IAVs and routinely used in influenza research [31] whereas the latter is the only commercially available equine-derived cell line. E.Derm cells are interferon-competent [32] and thus reflect better the intracellular context of an EIV infection in its natural host than MDCK cells. Fig 1A shows the number of mutations that separate EIV/63 from EIV/2003 as well as their nature (synonymous or nonsynonymous) and their distribution in each genomic segment. Given the importance of the hemagglutinin glycoprotein on virus host range and tropism, the amino acid mutations present in the HA gene that separate both viruses are listed in S1 Table. For clarity, S1 Fig shows the phylogenetic relationship between the viruses, which supports their monophyletic origin and is consistent with previous studies [27,33]. Upon infection of cells with EIV/63 and EIV/2003, we compared virus replication, cell-to-cell spread, and virus susceptibility to the type I interferon (IFN) response. In MDCK cells, we observed viral titres for both viruses increasing to similar levels between 4- and 8-hours post-infection (hpi). EIV/63 titres showed a sharp increase at 12 hpi, peaked at 24 hpi and steadily declined after this timepoint (Fig 1B). EIV/2003 titres increased more slowly in the first 12 hpi but continued increasing until 48 hpi, when they peaked at a titre three logs higher than EIV/63, which was maintained at 72 hpi (Fig 1B). However, the observed differences in growth kinetics were not significant (p-value= 0.575). To compare the ability of the viruses to spread from infected to surrounding cells, we performed immunofocus plaque assays in MDCK cells and measured the area of infectious foci at 48 hpi. EIV/2003-induced foci were significantly larger than those of EIV/63 (Fig 1C, p-value $< 10^{-17}$, student's t-test). Next, we compared the replication kinetics of the viruses in E. Derm cells. We observed significant differences (p-value = $7.98 \times 10^{-4}$) in virus growth: while both viruses displayed similar titres between 4 and 12 hpi, EIV/2003 grew to titres three logs higher than EIV/63 from 24 hpi onwards (Fig 1D). To test if the observed differences were due to distinct susceptibility of each virus to the cellular type I IFN response, we infected E.Derm cells with EIV/63 or EIV/2003 in the presence or absence of ruxolitinib, a JAK1/2 inhibitor that impairs the cellular response to IFN [34]. Interestingly, in the presence of ruxolitinib, EIV/63 replication was significantly higher (p-value = 0.039), reaching similar titres than those observed for EIV/2003 (p-value = 0.651, Fig 1D). In contrast, levels of EIV/2003 virus replication remained unaltered in the presence of ruxolitinib (p-value = 0.331), suggesting that this virus is not affected by the host type I IFN response. Overall, these results show that in mammalian cells EIV/2003 outperforms EIV/63 on the fitness components we measured (virus growth kinetics and cell to cell spread), and that EIV/2003 is less susceptible to the equine type I IFN response.

## EIV/63 and EIV/2003 elicit distinct transcriptomic responses in equine cells

The different phenotypes observed between EIV/63 and EIV/2003 in E.Derm cells led us to hypothesize that each virus triggers a distinct cellular response to infection. To identify these differences, we analysed the transcriptome profiles of E.Derm cells infected either with EIV/63 or EIV/2003 at 4 and 24 hpi. First, we analysed global transcriptomic changes by generating multidimensional scaling (MDS) plots. Fig 2A shows an overlap of transcriptomes of EIV-infected cells at 4 hpi, indicating similar cellular responses induced at early time post infection. The short distance observed in MDS plots between infected and control cells suggests that few transcriptomic changes take place in E.Derm cells early after infection. In contrast, at 24 hpi transcriptional changes induced by EIV/63 and EIV/2003 became separated from each other and from mock-infected cells, as observed by distinct, non-overlapping clusters of datapoints

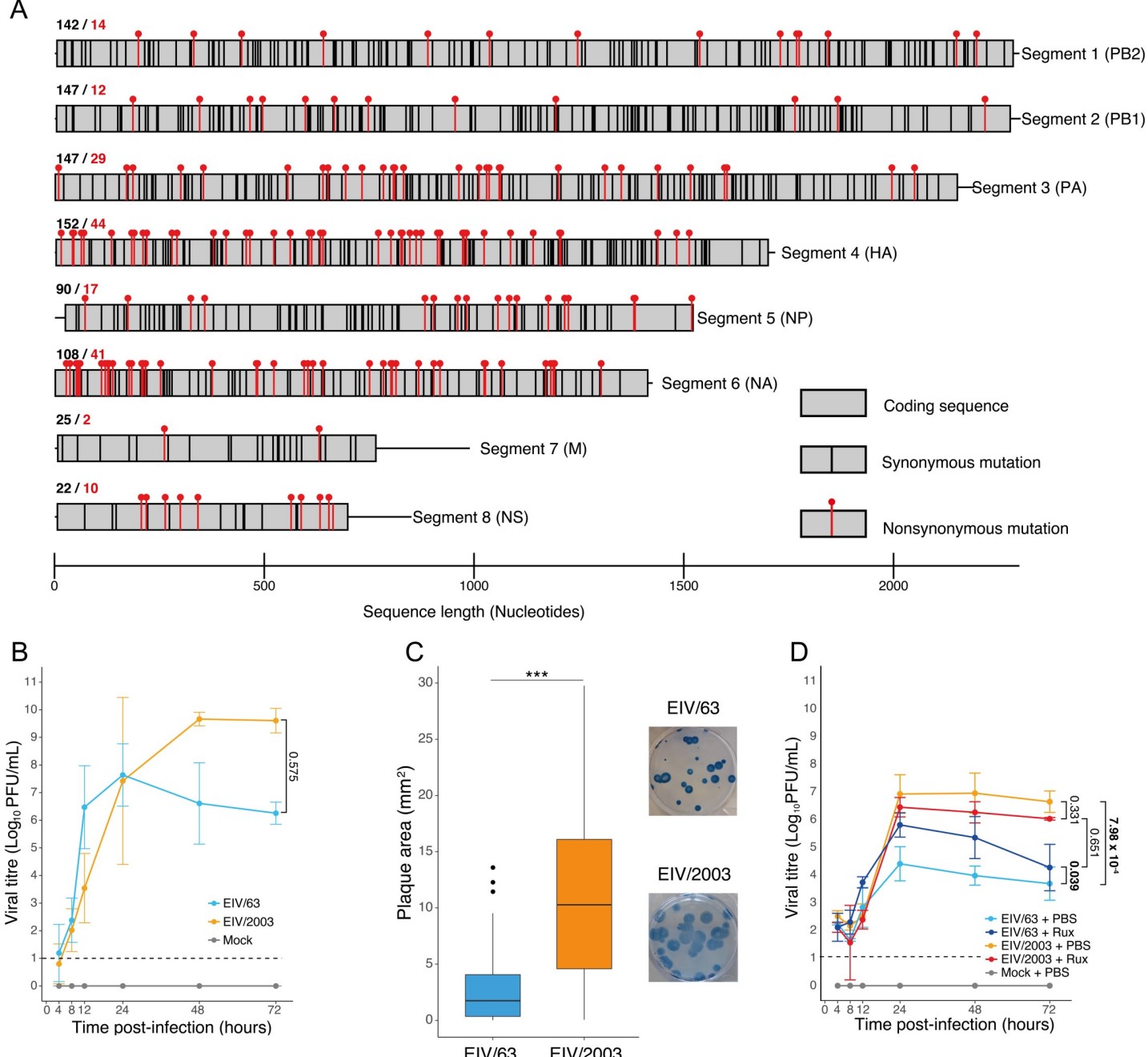

**Fig 1. *In vitro* characterisation of H3N8 EIVs.** (A) Comparison of EIV/63 and EIV/2003 coding sequences (CDS). The main protein CDS and associated polymorphisms among EIV/63 and EIV/2003 are shown as grey boxes. Black and red bars indicate synonymous and nonsynonymous mutations, respectively. The total number of synonymous and non-synonymous mutations per segment is shown above each segment in black and red, respectively. (B) Growth kinetics of EIV/63 and EIV/2003 in MDCK cells. (C) Plaque phenotype of EIV/63 and EIV/2003 in MDCK cells at 48 hpi. (D) Growth kinetics of EIV/63 and EIV/2003 in E.Derm cells with or without ruxolitinib treatment. Cyan and orange lines represent EIV/63 and EIV/2003 in the absence of ruxolitinib, respectively. Blue and red lines represent EIV/63 and EIV/2003 in the presence of ruxolitinib, respectively. Non-infected controls are shown in grey. Viral titres are represented as the mean ± SD. The limit of detection of plaque assays is shown as a black dashed line. Significant p-values are shown in bold and were calculated using generalized linear mixed-effects models. Infections were performed three times independently and each independent experiment consisted of three technical repeats. In panel C, 100 plaques per condition were measured and statistical significance assess using student t-test.

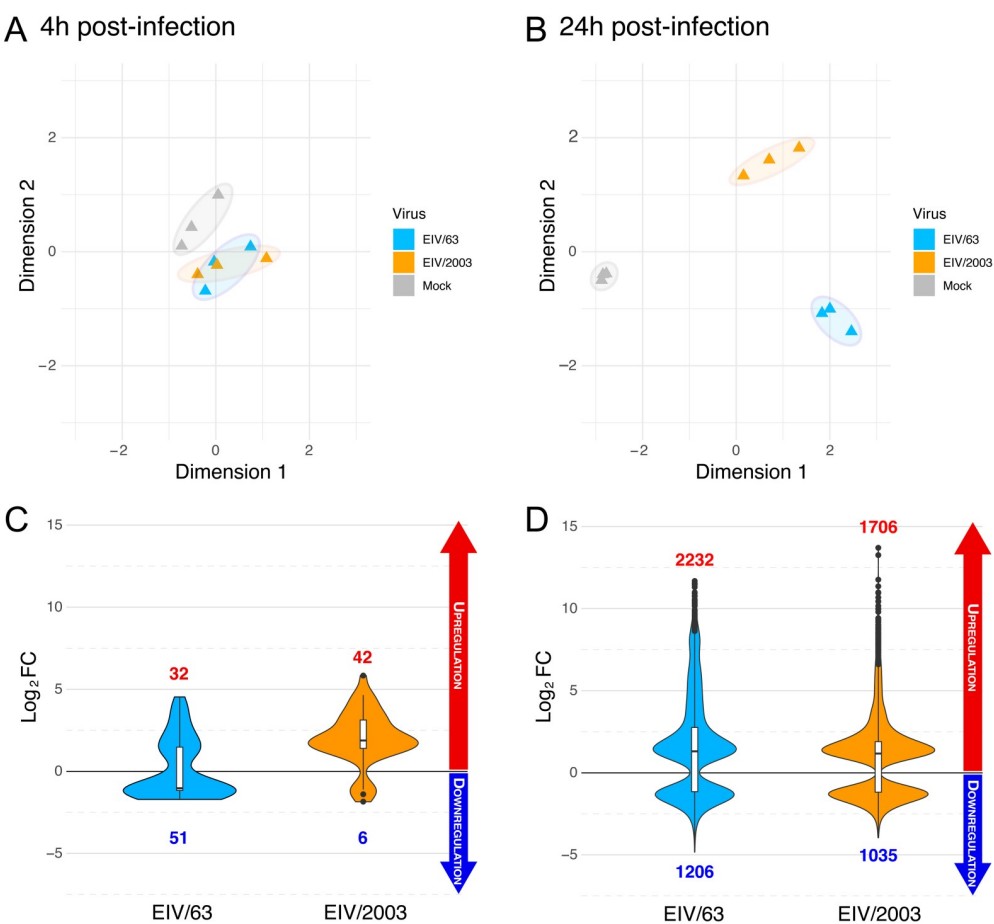

**Fig 2. Overview of EIVs transcriptomes in E.Derm cells.** (A-B) Multidimensional scale plots of transcriptomes of E. Derm cells infected with EIV/63 (cyan), EIV/2003 (orange), or mock-infected (grey) at 4 hpi (A) or 24 hpi (B). Interpoint distance correlates with divergence in transcriptomic profiles. (C-D) Violin plots of differentially expressed genes (DEGs) (p<0.05; |Log₂FC|>1) identified in E.Derm cells infected with EIV/63 (cyan) or EIV/2003 (orange) compared to mock-infected cells at 4 hpi (C) or 24 hpi (D). The number of upregulated and downregulated genes in each condition is shown in red and blue, respectively. Data presented here summarise three independent experiments.

reflecting each condition (i.e. mock-infected, EIV/63-infected, and EIV/2003-infected, Fig 2B). This result confirms that each virus induces time-dependent specific cellular responses. Furthermore, we quantified all differentially expressed genes (DEGs) and their respective fold change (Log₂FC) under each condition. At 4 hpi (Fig 2C), each virus induced less than a hundred DEGs in total: 83 for EIV/63 and 48 for EIV/2003. In addition, the net regulatory effects were markedly different as the majority of DEGs modulated by EIV/63 were downregulated (51/83, 61.4%), whereas the majority of DEGs modulated by EIV/2003 were upregulated (42/ 48, 87.5%). At 24 hpi, the number of DEGs was considerably higher than at 4 hpi (>2,700 for each virus). Again, EIV/63 displayed the largest number of DEGs compared to EIV/2003 (3438 vs. 2741, respectively, Fig 2D). The distribution of up/downregulated genes was similar at this timepoint, with 2232 (64.9%) and 1706 (62.2%) genes being upregulated, while 1206 and 1035 DEGs were downregulated by EIV/63 and EIV/2003, respectively (Fig 2D). We further identified the shared and non-shared DEGs induced by each virus. At 4 hpi, 28 genes were differentially expressed by both viruses, while 55 DEGs were exclusively modulated by EIV/63 and 20 only by EIV/2003 (Fig 3A). We compared the Log₂FC value (relative to mock-

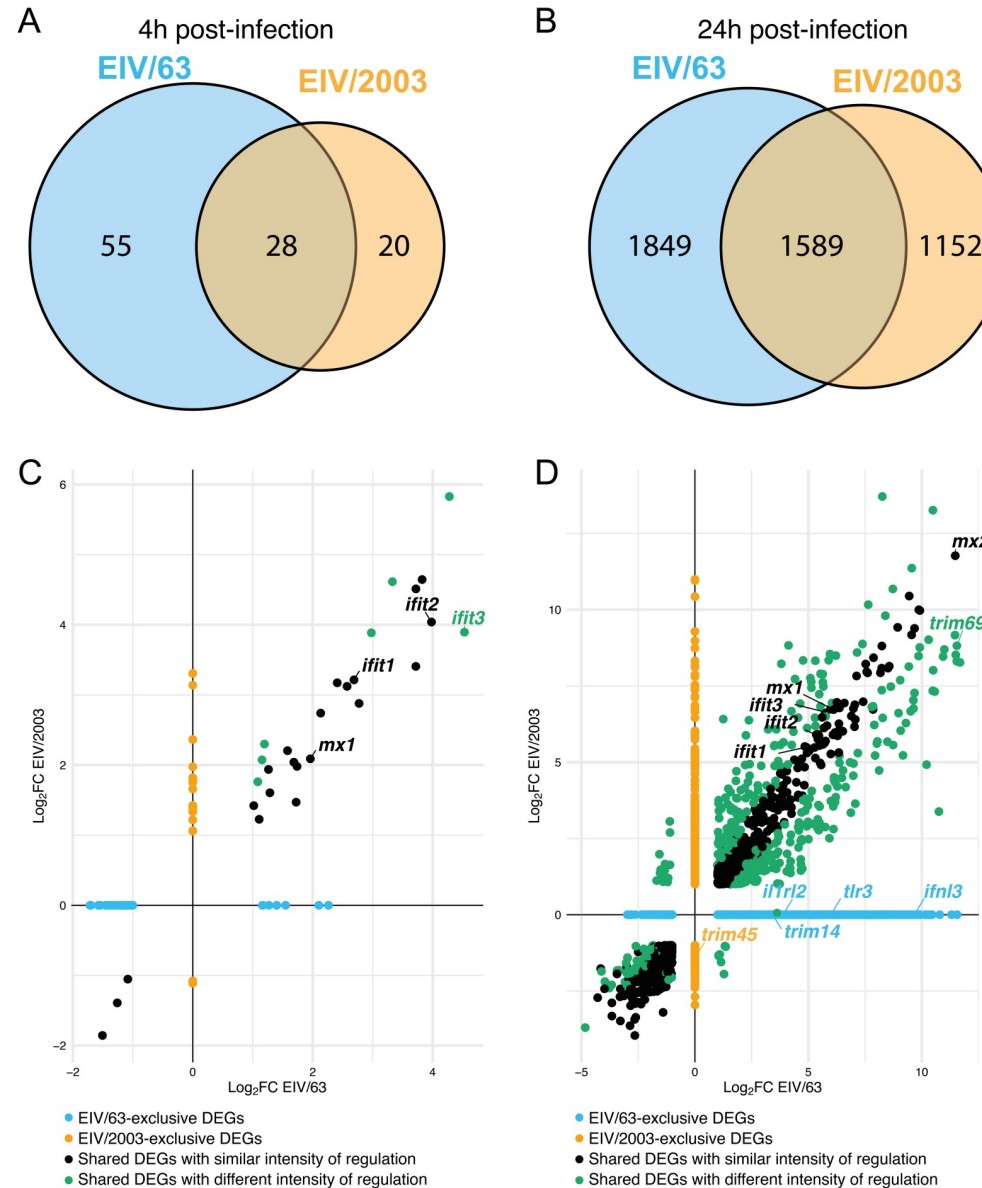

**Fig 3. Similarities and differences between the transcriptomes of EIV/63 and EIV/2003 in E.Derm cells.** (A-B) Distribution of differentially expressed genes (DEGs) for EIV/63 (cyan) and EIV/2003 (orange) at 4 hpi (A) and 24 hpi (B). The number of DEGs for each subset is shown within each compartment of the Venn diagram. (C-D) Pairwise comparison of the intensity of regulation ($Log_2FC$) for each DEG at 4 hpi (C), and 24 hpi (D). DEGs uniquely associated with EIV/63 or EIV/2003 are displayed in cyan, or orange, respectively, while common DEGs are coloured in black (non-significant) or emerald (significant). Data presented here represent three independent experiments.

infected cells) of the shared DEGs and observed that seven of them (green circles in Fig 3C) displayed statistically significant differences in their levels of dysregulation. Among these seven genes we identified *ifit3*, a well characterised innate immune activator [35], which showed higher upregulation following EIV/63 infection. Other well-characterized genes involved in the antiviral immune response (*mx1*, *ifit1*, *ifit2*) were found upregulated to similar levels by both viruses (Fig 3C). At 24 hpi, EIV/63 displayed 1849 (53.8%) unique DEGs and EIV/2003 1152 (42%), while the number of shared DEGs between both viruses was 1589 (Fig 3B). Of these 1589 genes, 490 showed significant differences in their transcription levels (green

circles in Fig 3D), indicating that the magnitude of their regulation was virus-specific. While the main innate immune regulators (*mx and ifit genes*) did not show significant transcriptional differences between the two viruses, we observed some important exceptions. For example, *trim69*, a gene associated to viral restriction [36], was upregulated at significantly higher level by EIV/63 (Fig 3D). Additionally, we observed a strong upregulation of *trim14*, *il1rl2*, *tlr3*, and *ifnl3* in EIV/63-only DEGs (Fig 3D). These genes are involved in the detection and innate immune defense against viruses, and are thought to play an important role in mucosal immunity (*il1rl2*, *ifnl3* [37]). This is particularly relevant as the respiratory mucosa (i.e. the lining of the respiratory tract) is the natural site of IAV infection in mammals. Furthermore, we observed that EIV/2003 -but not EIV/63- downregulated *trim45* (Fig 3D), which is associated with the interferon gamma signalling pathway, an important arm of the anti-viral response against IAV infection. Overall, these results show that the transcriptomes of equine cells infected by EIV/63 or EIV/2003 are quantitatively and qualitatively different and affect thousands of cellular genes. As the cellular environment remained the same (i.e. E.Derm cells), the observed differences must be due to genotypic differences between EIV/63 and EIV/2003.

## EIV/63 and EIV/2003 modulate the type I IFN-mediated innate immune response differently

Based on the results obtained in our transcriptomic experiments and the experimental infections using ruxolitinib, we hypothesized that EIV/63 and EIV/2003 modulate the type I IFN response differently. To test this, we compared the transcriptome of E.Derm cells following IFN treatment (i.e. the interferome) with the transcriptomes of EIV-infected cells. E.Derm cells were treated with universal type I IFN (uIFN, see materials and methods) and their transcriptomic profiles were determined at 24 hours post-treatment. Cells treated with uIFN displayed 143 DEGs (Fig 4), referred to as differentially expressed IFN-stimulated genes (DE-ISGs). Notably, only 87 (60.84%) of those 143 DE-ISGs were expressed in EIV-infected cells, highlighting the ability of H3N8 EIVs to limit the type I IFN response. In addition, 115 DE-ISGs showed similar EIV-induced regulation (relative to mock-infected cells) (Fig 4A), while 28 DE-ISGs exhibited distinct levels of transcription between the viruses (Fig 4B). Most of these 28 DE-ISGs are involved in pathogen sensing (i.e. *tlr3*, *nod2*, *aim2*), immune response (i.e. *tnfsf13b*, *cd274*), or are known cytokines (i.e. *cxcl8*, which encodes IL-8 and is important for neutrophil recruitment [38], lung damage [39], and defense against IAV infection [39]). We noted that genes leading to immune activation such as *tlr3* or *tnfsf13b* are significantly more upregulated in EIV/63 infected cells than in EIV/2003 infected cells (Fig 4B). We also observed that *cd274* (PD-L1, a protein known to prevent targeting of infected cells by CD8[+] T cells [40]), is upregulated at higher levels in EIV/2003 infected cells. Taken together, these results show that EIV/2003 avoids detection by innate immune sensors (e.g. *tlr3*, *nod2*, *aim2*), blocks T and B cell functions, which are essential for the control of IAV infection (e.g. *cd274*, *tnfsf13b*), and prevent upregulation of some well characterised ISGs (*gbp1*). In contrast, EIV/63 does not avoid detection by innate immune sensors (e.g. *tlr3*, *nod2*), and does not control expression of important genes involved in the adaptive immune response (e.g. *cd274*, *tnfsf13b*). However, EIV/63 specifically prevents upregulation of well-known ISGs (e.g. *irf9*, *rsad2*), but not *gbp1*.

## EIV/63 but not EIV/2003 induces transcriptomic changes associated with stronger inflammatory responses

Based on the differences observed in transcriptional changes induced by EIV/63 and EIV/2003, we expected the viruses to modulate biological processes differently. To test this, we

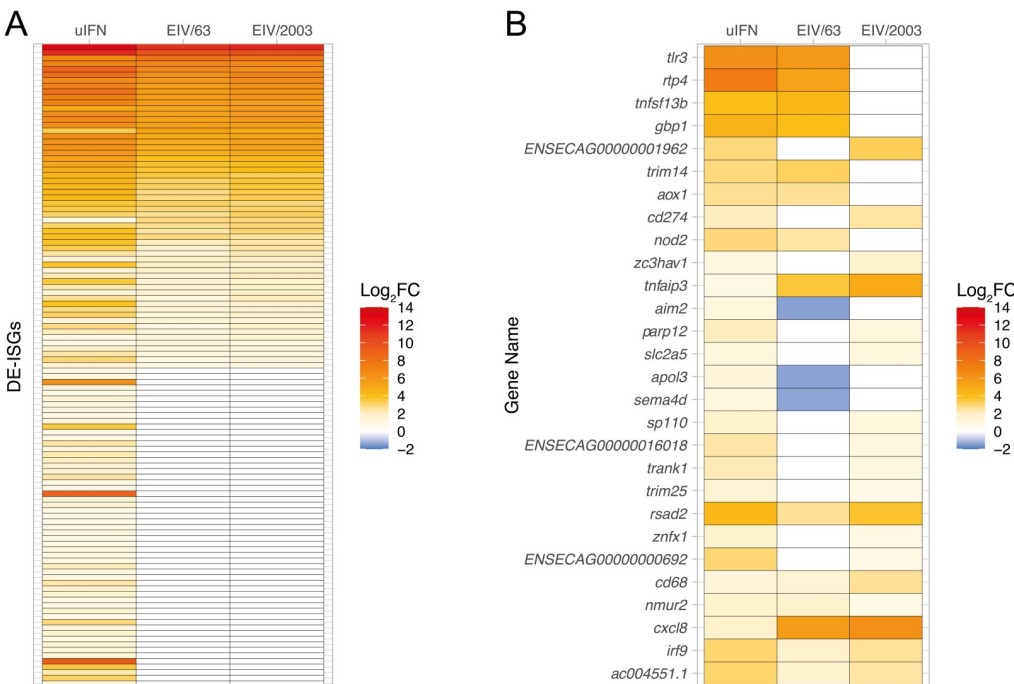

**Fig 4. Similarities and differences of ISG regulation induced by EIV/63 or EIV/2003 infections in E.Derm cells.** (A) Common differentially expressed interferon-stimulated genes (DE-ISGs) in E.Derm cells treated with universal type I IFN or infected with EIV/63 or EIV/2003 24 hpi. (B) DE-ISGs with significantly different levels of transcription between EIV/63 and EIV/2003 at 24 hpi. Gene expression levels are shown using a blue-white-red colour scale where blue to white indicates downregulation and white to red indicates upregulation. Data represents three independent experiments.

performed gene ontology (GO) enrichment on our transcriptomics data. Briefly, DEGs were first divided in subsets according to the viruses that induced them (i.e. DEGs induced by EIV/63 or EIV/2003) and by their regulatory effect (down/up regulation). At 4 hpi we observed 9 and 6 GO terms significantly enriched by EIV/63 and EIV/2003, respectively, of which 4 were shared by both viruses, and displayed similar gene ratio and regulation (S2 Fig). At 24 hpi we observed 103 and 98 GO terms significantly enriched by EIV/63 and EIV/2003, respectively, of which 31 were common to both viruses (S2 Fig).

As our previous results showed that EIV/63 and EIV/2003 modulate differently the type I IFN response, we focused on GO terms associated with host response to infection (as described in materials and methods). At 4 hpi, 4 of these GO terms were significantly enriched by both viruses (Fig 5A) and included *Defense response to virus*; *Type I Interferon signalling pathway*; *Response to virus*; and *Negative regulation of viral genome replication*. We noted that a unique GO term (i.e. *Response to type I* interferon) was present in EIV/63 but not in EIV/2003 infected cells at 4 hpi. At 24 hpi we observed the same shared GO terms that were enriched at 4 hpi, plus an additional one (*viral transcription*, Fig 5A). The shared GO terms exhibited similar regulatory effects (up/down regulation), number of genes involved, gene ratio, and significance (Fig 5A). Furthermore, we noted that 3 GO terms were exclusive to EIV/63 while other 4 different GO terms were unique to EIV/2003 (Fig 5A). The GO term *inflammatory response* was unique to EIV/63-infected cells and out of the GO terms that were exclusive for each virus it was the one that displayed the highest gene ratio, number of genes involved, and significance. We examined this GO term more closely, and found that out of 69 DEGs involved, 26 were common to both viruses (Fig 5B), 31 were unique to EIV/63, and 12

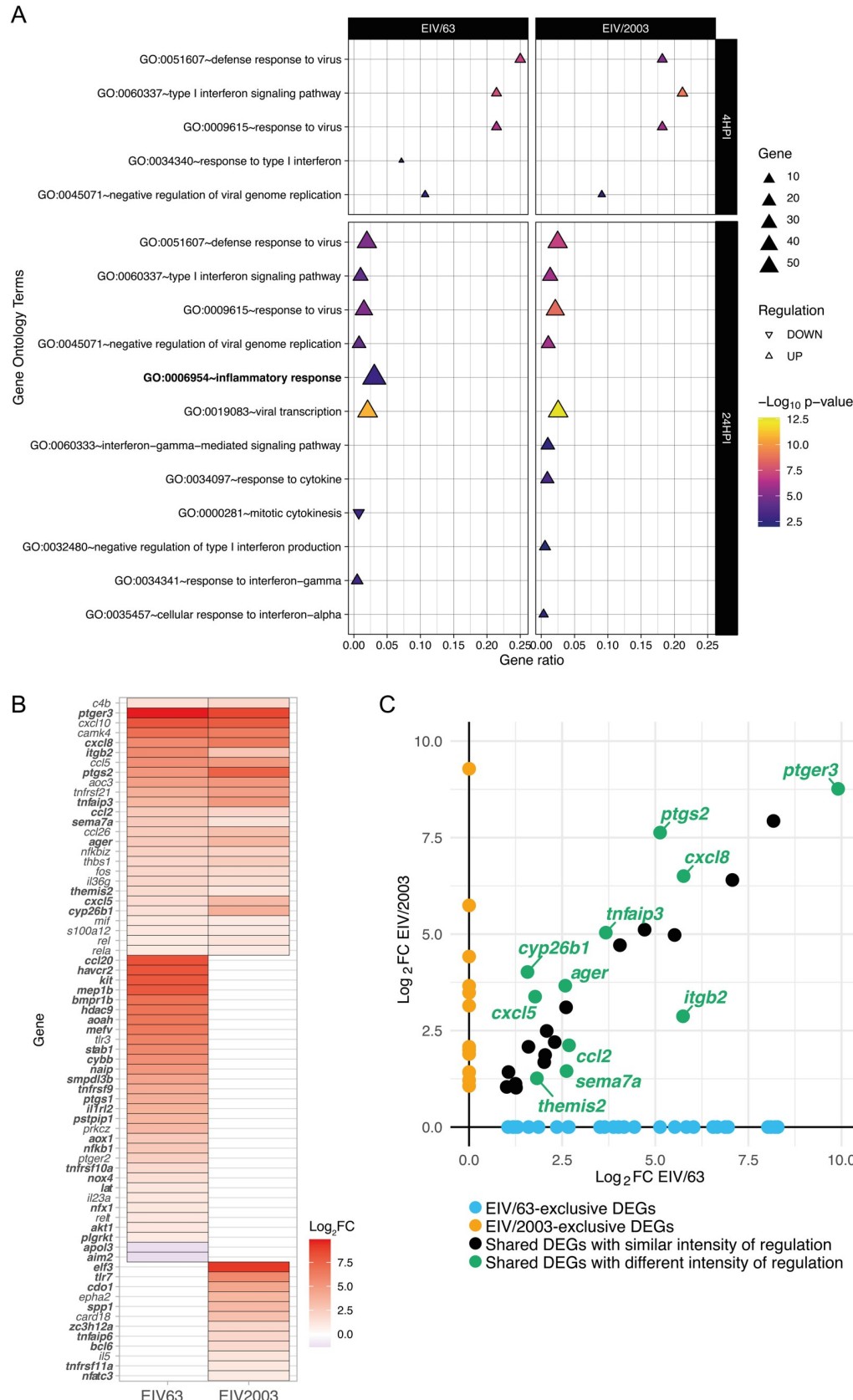

**Fig 5. Comparison of gene ontology (GO) terms associated to host response to infection in cells infected with EIV/63 or EIV/2003.** (A) GO terms associated to host response to infection (immunity or inflammation). Significantly upregulated GO terms are shown as triangles and significantly downregulated GO terms are shown as reverse triangles. The size of each triangle represents the number of DEGs fitting each GO term, while the ratio of total genes involved in each category is displayed on the x-axis. Triangles are coloured according to a blue-magenta-yellow gradient representing the enrichment significance (-Log$_{10}$ p-value). GO term associated with inflammation is shown in bold. (B) Regulation intensity at 24 hpi of significant DEGs involved in *GO:0006954 Inflammatory response* are shown in a heatmap, following a blue-white-red colour scale. Downregulation is highlighted by blue shades, while upregulation uses red shades. Significant DEGs among infectomes are shown with bold names. (C) Pairwise comparison of intensity of regulation (Log$_2$FC) and distribution of DEGs involved in *GO:0006954 Inflammatory response* at 24 hpi. DEGs uniquely associated with EIV/63 or EIV/2003 are displayed in cyan, or orange, respectively, while common DEGs are coloured in black (non-significant) or emerald (significant). Data presented here summarise three independent experiments.

to EIV/2003 (Fig 5B). All EIV/2003-exclusive DEGs were upregulated, and of the 31 unique DEGs identified in EIV/63 infected cells, 29 were upregulated and 2 downregulated (Fig 5B). All DEGs associated with this GO term and their respective transcription levels are shown in Fig 5C. Interestingly, 11 of the common DEGs were differently regulated by each virus (green circles in Fig 5C), and included *tnfaip3* (A20), a well-known negative regulator of NFκB activation and translocation [41], *ptgs2* (COX2), and *ptger3* (PGE2-R) genes which are both involved in the transformation of arachidonic acid into prostaglandin, a pro inflammatory molecule. Additionally, we observed the overexpression of *cxcl8*, *cxcl5*, and *ccl2* genes whose protein products are chemoattractant cytokines responsible for leucocyte recruitment (neutrophil, and monocyte), as well as *itgb2* gene which encodes LFA1 integrin, a protein playing an important role in the process of vascular extravasation of circulating leucocytes. Taken together, these results suggest that both viruses would trigger qualitatively similar local inflammatory responses, albeit of different magnitude (i.e. EIV/63 would trigger an exacerbated inflammatory response).

## EIV/2003 replicates and disseminates at higher levels in the equine respiratory tract but exhibits less tissue pathogenicity than EIV/63

Our results suggest that observed differences in fitness between EIV/63 and EIV/2003 are likely due to differences in the cellular response to each virus. To determine if such differences would also be present at the natural site of EIV infection, we infected *ex vivo*-cultured equine tracheal explants with EIV/63 and EIV/2003 and measured virus replication, intra-epithelial viral dissemination, and virus-induced histopathological changes. To measure virus replication, we titrated virus collected from the apical surface of infected explants at various times post infection (Fig 6A). EIV/2003 exhibited higher levels of virus replication (p-value = 0.0288, Fig 6A) consistent with our *in vitro* results in E. Derm cells. Virus intra-epithelial dissemination and histopathological changes were measured using image analyses of histological sections of formalin-fixed, paraffin-embedded equine tracheal explants collected at different times post-infection. To quantify virus dissemination, we performed immunohistochemistry using an antibody targeting the virus nucleoprotein (NP) and quantified the stained area on histological sections (Fig 6B). EIV/2003 infected explants showed statistically significant larger areas of NP staining than EIV/63 (p-value = 0.026), indicating increased virus dissemination (Fig 6C). Fig 7A shows representative images of sections of EIV-infected and mock-infected tracheal explants. To quantify virus-induced tissue injury, we measured the number of cells (Fig 7B) and the area (Fig 7C) of the epithelium. While the number of cells in the tracheal epithelium decreased steadily from day 1 to day 4 pi in EIV-infected explants and at a higher rate than in mock-infected explants (Fig 7B), the reduction in the number of epithelial cells was significantly more marked in explants infected with EIV/63 than EIV/2003 (p-value = 3.15x10$^{-3}$,

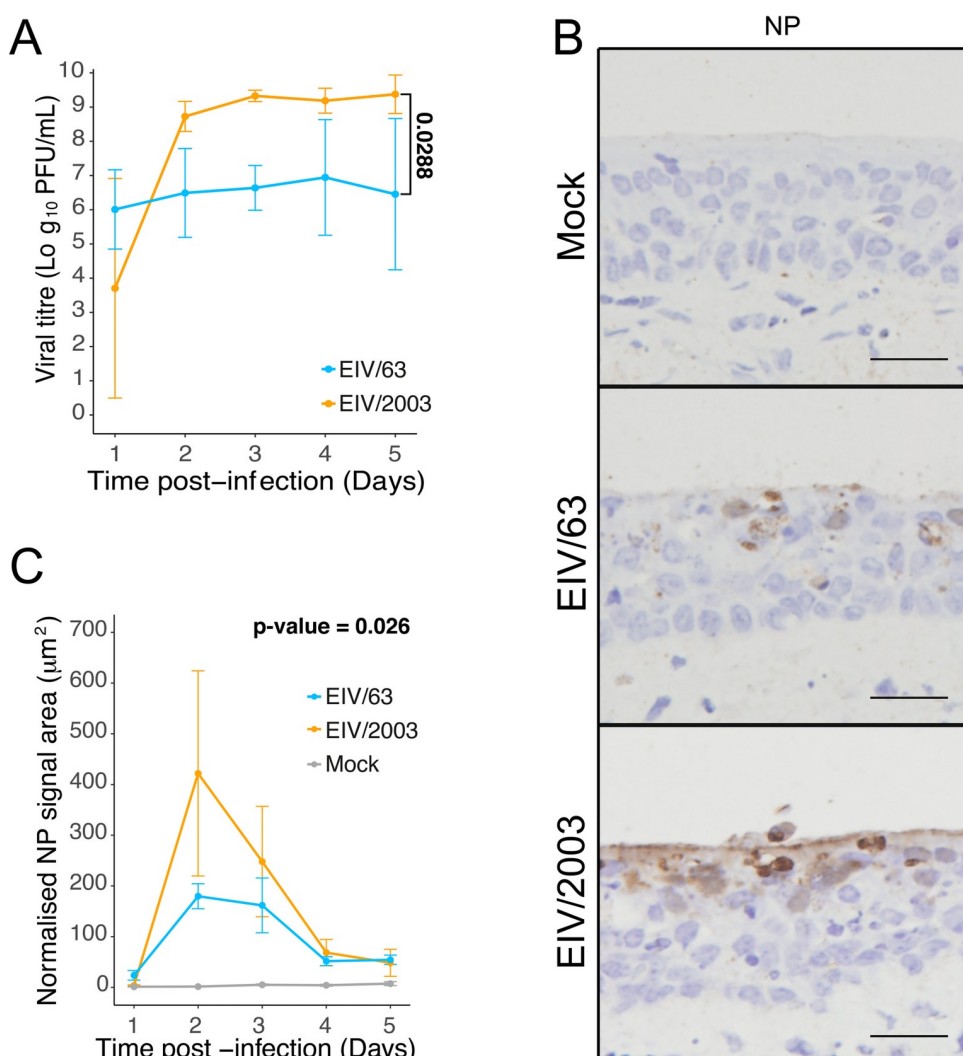

**Fig 6. Growth kinetics and tissue distribution of EIV/63 and EIV/2003 in equine tracheal explants.** (A) Growth kinetics of EIV/63 (cyan) and EIV/2003 (orange) in equine tracheal explants. Viral titres are represented as the mean ± SD of triplicate experiments. (B) Representative images of immunohistochemical staining using an anti-NP antibody in explants infected with EIV/63, EIV/2003 or mock-infected at day 2 post-infection. (C) Quantification of NP staining from day 1 to day 5 pi. Cyan, orange, and grey lines represent explants infected with EIV/63, EIV/2003 and mock-infected, respectively. Antigen quantification is summarised as the mean ± SEM of three images per section per timepoint on three independent sections. P-values are shown in bold when significant and were calculated using generalized linear mixed-effects models.

Fig 7B). Furthermore, we observed that the tracheal epithelium in EIV/63-infected explants was thinner when compared to EIV/2003-infected explants (p-value = $1.24 \times 10^{-3}$, Fig 7C), indicating that EIV/2003 caused less epithelial damage compared with EIV/63. It should be noted that the overall histopathological changes observed *ex vivo* were consistent with those found in horses experimentally infected with EIV [22], except for the inflammatory infiltration within the submucosa that could only be observed *in vivo*.

We also assessed cellular processes in response to viral infection. To compare the levels of virus-induced apoptosis, we quantified expression of cleaved caspase-3 (CC3), an effector caspase, which triggers the apoptotic process [42]. CC3 expression was higher in infected explants from day 1 to day 3 pi (Fig 7D), but on day 4 pi this trend was reverted. However, the overall

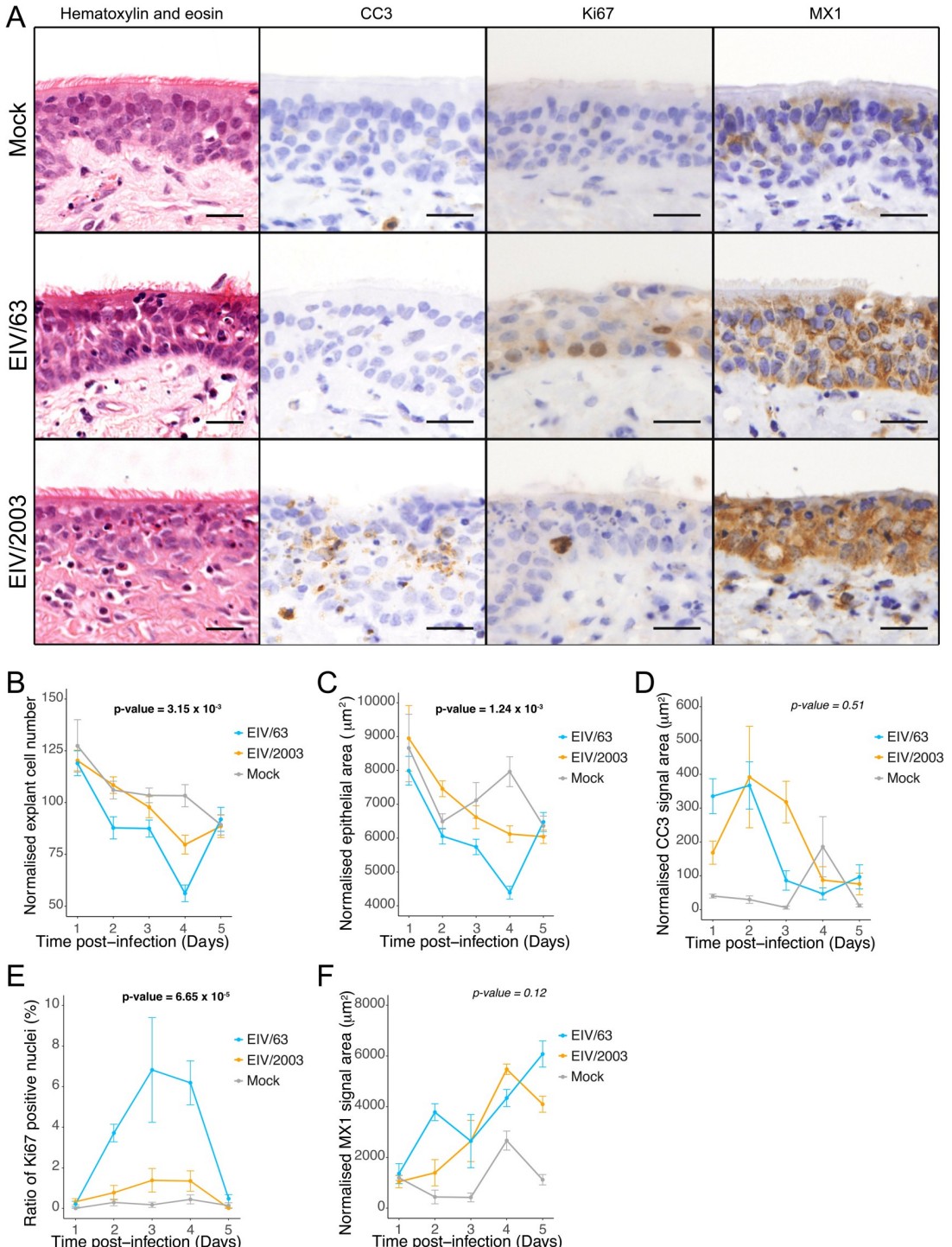

**Fig 7. Analyses of morphological changes and cellular processes (apoptosis, viral spread, tissue regeneration, and innate immune response) in equine tracheal explants infected with either EIV/63, EIV/2003, or mock-infected.** (A) Representative images of histological sections of explants infected with EIV/63, EIV/2003, or mock-infected at day 3 pi stained with haematoxylin and eosin (H&E), or immunostained (CC3, Ki67, MX1). (B-F) Plots showing measurements of the number of nuclei (B), epithelial area (C), cleaved-caspase 3 staining (D), Ki67 staining (E), and MX1 staining (F). Cyan, orange, and grey lines represent explants infected with EIV/63, EIV/2003 and mock-infected, respectively. Antigen quantifications are summarised as the mean ± SEM of three images per section per timepoint on three independent sections. P-values are shown in bold when significant and were calculated using generalized linear mixed-effects models.

kinetics of CC3 expression remained similar between EIV/63 and EIV/2003 infections (p-value = 0.51), but it appeared that EIV/2003-induced CC3 protein expression was delayed by approximatively 1 day when compared to EIV/63. To compare levels of epithelial regeneration (in response to cell death) we performed immunohistochemical staining targeting Ki67, a nuclear marker of dividing cells expressed in all active phases of the cell cycle [43]. Fig 7E shows that EIV/63-infected explants exhibited a significantly higher number of dividing cells than EIV/2003 (p-value = $6.65 \times 10^{-5}$), consistent with a stronger regenerative response to the high levels of EIV/63-induced cell death previously highlighted. As expected, non-infected explants displayed the lowest number of dividing cells. Finally, we compared the extent and kinetics of the innate immune response triggered by each virus by quantifying the expression of the ISG product MX1 (Fig 7F) [44]. Tissues infected with EIV/63 displayed two peaks of MX1 expression (at days 2 and 5 pi) whereas explants infected with EIV/2003 exhibited only one peak at day 4 pi, suggesting that the latter delays the expression of ISGs within the respiratory tract. However, the overall differences in MX1 expression over the five days post-infection were not statistically different (p-value = 0.12). Taken together, our results show that EIV/2003 replicates to higher levels and disseminates over larger areas within the equine tracheal epithelium, and while both viruses induce qualitatively similar pathological changes and virus-induced responses, the magnitude and kinetics of such changes are markedly different.

## Discussion

Understanding how changes in virus genotype due to long-term adaptation impact virus phenotype and fitness after pathogen emergence is unclear. Notably, most research work has focused on the effect of single mutations on specific functions of viral proteins contributing to individual traits, even though it is widely accepted that viral adaptation is a continuous and concerted process that affects multiple genes simultaneously. To address this issue, we examined several traits of the earliest (EIV/63) and a contemporary (EIV/2003) isolate of a monophyletic, avian-origin influenza virus lineage that has been continuously circulating in horses for almost 60 years. As we focused on within-host fitness (i.e. the ability of the viruses to colonise cells, spread in tissues, and produce infectious virions) we performed a series of experiments that enabled us to compare replication kinetics, cell-to-cell spread, susceptibility to type I interferon and respiratory epithelial injury.

To examine the effects of changes in virus genotype on replicative fitness -a key component of overall fitness [9]- we compared growth kinetics of EIV/63 and EIV/2003 in MDCK and E. Derm cells. While this experimental platform does not exactly recapitulate the process of natural selection undergone by the H3N8 EIV lineage in horse populations, it allowed us to compare in a quantitative fashion a particular feature of the virus replication cycle within cells and tissues that H3N8 EIVs can infect productively. Notably, we showed that EIV/2003 exhibits significantly higher replicative fitness relative to EIV/63 in E.Derm cells (Fig 1D) but not in canine-derived MDCK cells (Fig 1B), suggesting that mutations acquired during the evolutionary history of the H3N8 EIV lineage led to the adaptation of EIV to a horse-specific interferon-competent cellular environment. Our results are consistent with those observed by Stucker et al. [45], who showed the effect of multiple concerted mutations on the replicative fitness of canine parvovirus *in vitro*. While we cannot explain this effect mechanistically, our experiments using ruxolitinib (Fig 1D) and our transcriptomics experiments (Figs 3, 4 and 5) suggest that the type I IFN response is likely to play a central role in this phenotypic change.

To examine the evolution of H3N8 EIV adaptation in a more relevant biological context we used *ex vivo*-cultured equine tracheal explants. Explants and air-liquid interface (ALI) cultures of respiratory epithelium represent to a great extent the natural site of infection of respiratory

viruses and are broadly used to study the infection biology of many viruses including IAVs, severe acute respiratory syndrome coronavirus 2 (SARS-CoV-2), and respiratory syncytial virus (RSV) [46–48]. Regarding IAVs, back to back infections of pigs and ALI-cultures derived from pigs with different isolates of pdmH1N1 showed that the infection phenotype of ALI-cultures can be used to predict *in vivo* virulence [49]. Importantly, as explants lack any component of adaptive immunity, they allow to evaluate virus-cell interactions at the natural site of infection, excluding systemic responses and/or fitness effects resulting from antigenic drift. Virus titrations and image analysis of infected explants show that EIV/2003 replicates to higher levels (Fig 6A) and spreads over larger areas of the respiratory epithelium (Fig 6C) than EIV/63, suggesting that genotype changes acquired during H3N8 EIV evolution resulted in higher pathogen load.

Virulence, defined as a reduction in the fitness of the host due to infection, can be measured in terms of disease severity or pathogen-induced death, and both are the result of within-host interactions that include virus-induced lesions in cells and tissues (such as lysis and transformation), and the host response to infection (e.g. apoptosis and inflammation). Severe and lethal outcomes are often linked to immunopathological responses mediated by so called *cytokine storms* [50]. As an evolvable trait, virulence is subject to natural selection and thus contributes to pathogen fitness. However, whether virulence is adaptive or not has been subject of debate [51]. To determine if EIV evolution was associated with changes in tissue damage we quantified lesions induced by each virus in respiratory explants using image analyses. EIV/63-infected explants exhibited higher extent of lesions, measured as a decrease of the respiratory epithelium area (Fig 7C), most likely due to a higher reduction in the number of epithelial cells (Fig 7B). While our explant system does not allow us to examine histological changes due to inflammation, our transcriptomics analyses of infected equine cells indicate that EIV/63 triggers a cellular response associated with stronger inflammatory processes than EIV/2003, especially at 24 hpi (Fig 5). Overall, these results suggest that H3N8 EIV evolved towards lower virulence and are in line with a previous report that compared the *in vivo* virulence of two H3N8 EIVs separated by 5 years of evolution (Sussex/89 [EIV/89] and Newmarket/93 [EIV/93]). In this study, Wattrang et al. showed that horses infected with EIV/93 (i.e. the virus likely to be better adapted based on the time it circulated in horse populations) exhibited milder disease -measured in terms of coughing, nasal discharge and rectal temperature- and lower IFN and IL-6 responses than those infected with EIV/89 [52]. We note that the decrease in tissue damage is not necessarily a shift towards *avirulence*, but more likely a trend towards an intermediate level of virulence as observed by other viruses such as myxoma virus in rabbits [53] and Newcastle disease virus in wild birds [54].

Our transcriptomics findings provided insight on the virus-host interactions that are likely to play an important role on virus adaptation. For example, the overall reduction in transcriptional changes induced by EIV/2003 when compared with EIV/63 is consistent with a previous study showing that in human cells, H3N2 human influenza virus dysregulated a lower number of genes than avian-origin H5N1 and H7N7 at 24 hpi [55], suggesting that adapted viruses evolve towards higher transcriptional efficiency (i.e. they disrupt a lower number of cellular genes while modifying a higher number of cellular pathways). Further, our transcriptomics results provide biologically plausible explanations to the observed changes in within-host fitness. As mentioned above, the different susceptibility of EIV/63 and EIV/2003 to the type I IFN response (Fig 1D) together with the distinct transcriptional changes induced by each virus on interferon-stimulated genes (ISGs) (Fig 4A and B) indicate that H3N8 EIV adaptation led to a more efficient escape to host pathogen recognition, as well as a more regulated control of the innate immune response. This is particularly important when considering that the innate immune response is the first line of defence against infections as it has direct antiviral effects

[56], modulates downstream immune responses [57], and possesses proinflammatory properties [58].

Overall, our study shows compelling evidence that H3N8 EIV evolution resulted in higher within-host fitness. Given the level of divergence between EIV/63 and EIV/2003 (Fig 1A), performing experiments using site-directed mutagenesis to determine the impact of individual amino acid mutations in the observed fitness differences is unfeasible not only because of the large number of mutants to be tested, but also due to the number of combinations that could increase fitness by means of epistatic effects. In addition, synonymous mutations are likely to play an important role in EIV adaptation: the distribution of CpG dinucleotides in IAV genomes mimics that of the host [59]. It has been shown that CpG frequency in H3N8 EIVs decreased steadily over time [60], consistent with continuous adaptive evolution. S3 Fig shows a significant and continuous reduction in CG and CpG frequency in H3N8 EIVs genomes isolated between 1963 and 2003 (p-value = $2x10^{-16}$, and $2x10^{-15}$, respectively). This reduction is likely to result in an enhanced innate immune evasion by minimising pathogen recognition [59,61].

In sum, our results suggest that long-term evolution following the host shift of an avian-origin influenza virus to horses resulted in the selection of multiple traits including higher growth rates and intra-host dissemination, as well as reduced tissue damage. Our work provides new insights on a central aspect of virus adaptation following emergence and highlights the need for context dependency in evolutionary theory.

## Materials and methods

### Ethics statement

Animal work was approved by the Ethics Committee of the School of Veterinary Medicine of the University of Glasgow (ethics approval R25A/12). As no regulated procedures were carried out on animals a Home Office license was not required.

### Cells

Madin-Darby Canine Kidney (MDCK; ATCC CCL-34) and 293T cells (ATCC CRL-3216) were grown at 37°C and 5% $CO_2$ in Dulbecco's modified Eagle's medium (DMEM) high glucose, GlutaMax, and pyruvate (ThermoFisher Scientific) supplemented with 10% Fetal Bovine Serum (FBS; Gibco Life Technologies). E.Derm cells (ATCC CCL-57) were grown at 37°C and 5% $CO_2$ in DMEM high glucose, GlutaMax, and pyruvate supplemented with 15% FBS, and 1% nonessential amino acids (NEAA; Gibco Life Technologies).

### Tracheal explants

Horse tracheas were aseptically collected upon euthanasia from healthy Welsh Mountain ponies and prepared as previously described [62].

### Plasmids

Plasmids sets used in this study are ambisense pDP-2002 plasmids carrying individual genomic segments of A/equine/Uruguay/1963 (EIV/63, [63]), or A/equine/Ohio/2003 (EIV/2003; [64]). Every set was composed of 8 individual plasmids corresponding to the 8-segmented viral genome.

## Viruses

Virus stocks were generated by reverse genetics as previously described [30,64]. Briefly, a co-culture of 293T and MDCK cells was cotransfected using TransIT-LT1 (Cambridge Bioscience) and 8-plasmids corresponding to the 8-genomic segments of either EIV/63 or EIV/2003. The day after, transfection cocktails were replaced by infectious medium (DMEM high glucose, GlutaMax, and pyruvate supplemented with 0.3% Bovine Serum Albumin (BSA; Gibco, Life Technologies), and 1μg/mL of tosylsulfonyl phenylalanyl chloromethyl ketone (TPCK)-treated trypsin (Sigma-Aldrich). Two to three days later, viruses were harvested and grown by infection of freshly plated MDCK cells (MOI 0.01) and stored at -80˚C.

## Virus titrations

Virus stock titres were calculated by standard plaque assay in MDCK, and E.Derm cells [30,64]. Briefly, MDCK, or E.Derm cells were infected by serial 10-fold dilutions. Following 1-hour incubation, cells were washed with sterile phosphate-buffered saline (PBS) and incubated with a mix of 2X Temin's modification MEM (ThermoFisher Scientific), 1.2% Avicel (FMC BioPolymer), and 1μg/mL of TPCK for 48-hours. TPCK was only added to MDCK cells. Following fixation for 10' with cold 80% acetone, cells were permeabilised using 1% TritonX-100 (Sigma-Aldrich) before being blocked in 10% Normal Goat Serum (NGS; Gibco, Life Technologies) and incubated overnight at 4˚C with primary mouse antibody anti Influenza (NP subtype A) clone EVS238 (1/1500; European Veterinary Labs). The following day, cells were incubated with secondary anti-mouse IgG, HRP-linked (1/3000; Cell Signaling) antibody, followed by the addition of TrueBlue peroxidase substrate (Insight Biotechnology). The limit of detection of the assays under our conditions was 10 PFU. Viral titres were expressed as $Log_{10}$ PFU/mL. Viral titres below the limit of detection were given a value of zero.

## Phylogenetic and genome composition analysis

We downloaded 136 complete H3N8 Equine Influenza virus sequences from the NCBI Influenza Virus Resource Database. Nucleotide alignments of the main coding sequences (CDS) were generated for each segment using MUSCLE [65]. We used RAxML [66] to infer the phylogenetic tree for each segment using the GAMMA model and 1000 bootstrap replicates. Trees were edited using FigTree v1.4.4 software (https://github.com/rambaut/figtree/releases). We calculated the number of occurrences of adenine (A), cytosine (C), guanine (G), and thymine (T) as well as CG dinucleotide (CpG), in each of the 8 segments using the SSE v1.4 software [67].

## Experimental infections

Virus growth kinetics infections were performed at a MOI 0.1 in MDCK and E.Derm cells. Infections were done using infectious medium or infection medium deprived of TPCK (E. Derm infection medium), respectively. Mock infections were carried out by treating both cell lines with their respective infection medium only. Interferon (IFN) treatment of E.Derm cells was as follows: cells were treated with 500 units of universal type I interferon (uIFN, pbl Assay Science) in infection medium and maintained during the whole-time course of the experiment. Equine tracheal explants were infected by adding 200 PFU of each virus in a final volume of 5uL on the epithelium surface. A similar volume of infection medium only was deposited on the surface of mock-infected explants. To measure virus growth kinetics in the absence of type I IFN response, E.Derm cells were treated with S-Ruxolitinib (Selleck Chemicals) at a concentration of 4 μM. Treatment started 24-hours prior to infection and was

maintained at the same concentration during the whole experiment. Supernatants from infected MDCK or E.Derm, as well as media from tracheal explant cultures were collected at various times post-infection, and titrated in MDCK cells by plaque assay. For transcriptomic experiments, E.Derm cells were infected by serial 2-fold dilutions of virus in E.Derm infectious medium, and samples were collected after 4 or 24-hours incubation. Infections (including Mock-infections) were performed three times independently and each independent experiment consisted of three technical repeats.

## Flow cytometry

Flow cytometry was used to determine the proportion of infected cells (E.Derm cells) for transcriptomics analyses. At 4-hours post-infection (hpi), cells were detached with trypsin-EDTA (Gibco; Life Technologies) and fixed in 4% freshly made Formalin solution (Sigma-Aldrich). Cells were permeabilised (1% TritonX-100), blocked (10% NGS), immunostained with Mouse anti Influenza (NP subtype A, clone EVS238, European Veterinary Labs, at a dilution of 1/1500), and with Rabbit anti-Mouse IgG conjugated with Alexa Fluor 488 (1/2000; Thermo-Fisher Scientific). Stained cells were subject to flow cytometry analysis (Guava flow cytometer; Merck), and the percentage of infected cells was calculated using FlowJo v10.3 software. The gating strategy used to identify viable and infected cells populations is shown in S4 Fig. Infections reaching 50% ± 5% infected cells at 4 hpi and their corresponding dilution at 24 hpi were RNA-extracted. Experiments were carried out three times independently.

## RNA sequencing

Total RNA was extracted using TRIzol reagent (ThermoFisher Scientific) and further purified according to the manufacturer's protocol using RNeasy mini-spin columns (Qiagen). Sample RNA concentration was measured with a Qubit Fluorimeter (Life Technologies) and the RNA integrity was determined using an Agilent 4200 TapeStation. 500 ng of total RNA from each sample were used to prepare libraries for sequencing, using an Illumina TruSeq Stranded mRNA HT kit, according to the manufacturer's instructions. Briefly, polyadenylated RNA molecules were captured, followed by fragmentation. RNA fragments were reverse transcribed and converted to dsDNA, end repaired, A-tailed, ligated to indexed adaptors and PCR amplified. Libraries were pooled in equimolar concentrations and sequenced in an Illumina NextSeq 500 sequencer using a high output cartridge, generating single reads with a length of 75 bp. The sequence reads are publicly available (BioProject accession number PRJEB29313).

## Transcriptome analyses

The quality of raw reads was assessed using FastQC [68]. High quality reads were mapped to the horse genome (Equus caballus, accession number GCA_000002305.1) using TopHat2 [69] and Bowtie2 [70]. The number of aligned reads were counted using HTseq [71] and normalised to counts per million (CPM). A list of differentially expressed genes (DEGs) compared to mock-infected samples was generated using EdgeR [72]. Genes with Benjamini-Hochberg corrected P-values < 0.05 were considered significant, and an additive expression threshold of $Log_2FC > |1|$ applied. To investigate the biological roles of these DEGs, functional annotation, and more precisely Gene Ontology terms (GO terms) were investigated using the Database for Annotation, Visualisation and Integrated Discovery [73]. We performed a selection of significantly enriched GO terms in relation with immune and/or inflammation responses based on the presence of the following terms in the GO name: 'interferon', 'IFN', 'cytokine', 'virus', 'viral', 'ISG', 'inflam' and 'immun'. The code used to perform transcriptome analyses is available in S1 File.

## Histological staining and immunostaining

Explants were fixed in 10% (v/v) buffered formalin and paraffin embedded. Multiple serial sections were obtained from each explant. Sections were stained with Haematoxylin and Eosin (H&E) and immunostained using various antibodies, including rabbit polyclonal anti Influenza NP (European Veterinary Labs), rabbit monoclonal anti-Cleaved Caspase 3 (D175; Cell Signalling), mouse monoclonal anti-MX1 (clone M143, provided by Georg Kochs), or mouse monoclonal anti-Ki67 (MIB-1; Dako). Endogenous peroxidase was blocked with 10% hydrogen peroxide + 0.05% Tween 20 (Scientific Laboratory Supplies). For the pre-treatment, slides were incubated with 1% TritonX-100 for NP staining, or pressure-cooked (Menarini Diagnostic) in pH 6.0 citrate buffered solution for the remaining primary antibodies (i.e. CC3, Ki67 and MX1). After a blocking step (10% Normal Goat Serum + 0.05% Tween 20), slides were incubated with primary antibody at the following dilutions: anti-NP 1:400, anti-Cleaved Caspase 3 1:400, anti-MX1 1:200, and anti-Ki67 1:150. Mouse EnVision/HRP (Dako, Agilent) or Rabbit EnVision/HRP (Dako, Agilent) were used as secondary antibodies. Liquid DAB (3.3'-Diaminobenzidine) + Substrate Chromogen System (Dako, Agilent) was used as chromogen. Sections were finally counterstained with Mayer's haematoxylin, dehydrated in ascending alcohol series and xylene, and coverslipped. Images were taken at 40x magnification with an Olympus BX51 microscope. Three images per section per timepoint were acquired and repeated for each staining. Additionally, every staining was measured on three sections.

## Image analyses

Epithelium areas were extracted from the surrounding tissues by manual contouring, before being measured (length and area). Images were colour deconvoluted [74] to separate brown (DAB) to blue (nuclei) channels, and default automatic thresholding was used to identify areas of positive signal. Binarisation of the DAB vector allowed calculation of positive pixels, which were then converted into area of positive stain for each marker. For nuclear stains, the image calculator was used to determine the localization of the positive pixels. All image analyses were done using ImageJ software (v2.0.0-rc-56/1.52g).

## Statistical analyses

Data visualisation and statistical analyses were carried out using R studio v3.5.1 software. Generalized linear mixed-effects models were implemented using the lme4 package [75] to investigate statistical associations between: i) viral titre and virus, considering replicates as a random effect in viral replication experiments, ii) staining area with virus. The goodness of fit of the models was assessed using the DHARMa package [76] version 0.4.1. Statistical differences in plaque size were calculated using a student's t-test analysis. GC content and CpG observed/expected (O:E) ratio correlation to virus sampling date were investigated using Pearson correlation test. Significance levels used are $^* < 0.05$; $^{**} < 0.01$; $^{***} < 0.001$. Graphs were generated using ggplot2 [77] and VennDiagram [78] packages. Error bars were calculated using mean value and standard deviation (SD) for viral growth curves, while standard error of the mean (SEM) was used for immunohistochemistry quantifications.

The numerical data used in all figures are included in S1 Data.

## Supporting information

**S1 Fig. Phylogenetic relationship of H3N8 EIVs isolated between 1963 and 2003.** Maximum likelihood trees using 136 complete H3N8 EIV genomes. Each tree represents the phylogenetic relationship inferred for each of the eight viral genomic segments. The name of each genomic

segment is indicated as follows: PB2 (polymerase basic 2); PB1 (polymerase basic 1); PA (polymerase acidic); HA (hemagglutinin); NP (nucleoprotein); NA (neuraminidase); M (matrix); and NS (non-structural). EIV/63 is indicated in cyan and EIV/2003 in orange. Nodes supported by a bootstrap value $\geq 75$ are shown in red.
(TIF)

**S2 Fig. Comparison of gene ontology (GO) terms in cells infected with EIV/63 or EIV/ 2003.** Significantly upregulated and downregulated GO terms are displayed as triangles or reverse triangles, respectively. The size of each triangle represents the number of DEGs fitting the specified GO term, while the ratio of total genes involved in each category is displayed on the x-axis. Triangles are coloured according to a blue-red-yellow gradient representing the enrichment significance ($-\text{Log}_{10}$ p-value).
(TIF)

**S3 Fig. Changes in viral genome composition along the evolutionary history of H3N8 EIV from 1963 to 2003.** The top and bottom scatter plots show the evolution of CG content, and CpG observed/expected ratio (CpG O:E Ratio), respectively, calculated from 136 H3N8 EIV complete genomes isolated between 1963 to 2003. Linear regressions between virus isolation and either CG content or CpG O:E ratio are shown in red, while 95% CI are shown in grey. P-values and $R^2$ are shown in red for each scatter plot.
(TIF)

**S4 Fig. Gating strategy used to analyse EIV-infected E.Derm in flow cytometry experiment.** The scatter plots and density plot show from left to right the gates used to distinguish viable E. Derm (FSC-H/SSC-H), singlet (SCC-A/SSC-H), and GFP$^+$ cells in mock-infected condition. The percentage on gated cells is given on each of the three plots.
(TIF)

**S1 Table. List of nonsynonymous mutations in segment 4 (HA) that differentiate EIV/63 from EIV/2003.** Nucleotide and amino acid positions are listed.
(TIF)

**S1 Data. Excel spreadsheet containing, the numerical data for Figs 1B, 1C, 1D, 6A, 6C, 7B, 7C, 7D and 7E.** Data for each figure is presented in a separate.
(XLSX)

**S1 File. Text file with the R Code use to analyse RNAseq data.**
(TEXT)

## Acknowledgments

We thank Lynn Stevenson, Lynn Oxford and Frazer Bell for technical assistance, and Daniel Streicker and Joanne Haney for providing feedback on the original manuscript.

## Author Contributions

**Conceptualization:** Luis Martinez-Sobrido, Pablo R. Murcia.

**Data curation:** Quan Gu.

**Formal analysis:** Julien A. R. Amat.

**Funding acquisition:** Julien A. R. Amat, Pablo R. Murcia.

**Investigation:** Julien A. R. Amat, Veronica Patton, Caroline Chauché, Joanna Crispell, Alice M. Coburn, Gaelle Gonzalez, Daniel Mair, Lily Tong, John F. Marshall.

**Methodology:** Daniel Goldfarb.

**Project administration:** Pablo R. Murcia.

**Supervision:** Francesco Marchesi, Pablo R. Murcia.

**Visualization:** Julien A. R. Amat.

**Writing – original draft:** Julien A. R. Amat, Pablo R. Murcia.

**Writing – review & editing:** Veronica Patton, Caroline Chauché, Daniel Goldfarb, Joanna Crispell, Quan Gu, Alice M. Coburn, Gaelle Gonzalez, Daniel Mair, Lily Tong, Luis Martinez-Sobrido, John F. Marshall, Francesco Marchesi.

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
