## [Decision Letter · Decision Letter 0]

30 Nov 2021

Dear Prof. Murcia,

Thank you very much for submitting your manuscript "Long-term adaptation following influenza A virus host shifts results in increased within-host viral fitness due to higher replication rates, broader dissemination within the respiratory epithelium and reduced tissue damage" for consideration at PLOS Pathogens. Your manuscript was reviewed by members of the editorial board and by the reviewers of the original submission. Based on the reviews, we are likely to accept this manuscript for publication, providing that you make further minor modifications as recommended.

Sincerely,

Yvonne Su, Ph.D.

Guest Editor

PLOS Pathogens

Marco Vignuzzi

Section Editor

PLOS Pathogens

Kasturi Haldar

Editor-in-Chief

PLOS Pathogens

orcid.org/0000-0001-5065-158X

Michael Malim

Editor-in-Chief

PLOS Pathogens

orcid.org/0000-0002-7699-2064

Reviewer Comments (if any, and for reference):

Reviewer's Responses to Questions

**Part I - Summary**

Reviewer #1: In this resubmission, the authors have made extensive textual edits throughout this manuscript, and have contributed additional information that was lacking in the original submission. These changes have improved the quality and utility of this study. However, there are still areas in the manuscript which warrant further improvement.

**Part II – Major Issues: Key Experiments Required for Acceptance**

Reviewer #1: Inclusion of a new Figure 1A is helpful to understand the magnitude of amino acid differences present between the two viruses under study, but is still lacking specificity. Considering the high number of nonsynonymous changes in the HA between the two viruses (including in HA1, which could impact receptor binding specificity), it would be beneficial if the authors provided a supplemental table to disclose exactly which amino acid changes are present in this protein between the two viruses. In PMID 21430049 (which shares authors with this study), several sites in the HA, including those linked with antigenic sites, were identified between H3N8 equine viruses over time; providing a list of amino acid differences in the HA for the two viruses under investigation here would thus be very helpful to the reader.

**Part III – Minor Issues: Editorial and Data Presentation Modifications**

Reviewer #1: 1. Lines 52 and 135: authors state they are examining differences in “infection phenotype” but are employing replication-based assays in Figure 1B and 1D to assess this; wording should be changed to replication so not to mislead the reader, and to better match how the authors describe these investigations on line 168 (“virus growth kinetics and cell to cell spread”).

2. Lines 451-454: the authors state that the plasmids for EIV/63 are derived from reference 63, and the plasmids for EIV/2003 are derived from reference 64; however, it appears that reference 63 is incomplete (is this a dissertation? Book chapter?) so the reader cannot follow what the authors are conveying, and reference 64 only mentions plasmids from EIV/63, not EIV/2003. Please ensure both of these references are correct and accurately link to the information intended.

3. The authors state in their response to reviewer comments that the limit of detection for plaque assays was 10 PFU, but did not include this information in the methods section; please add this. In this vein, Figures 1B and 1D show mock-infected cells had titers of 0 PFU/ml which would be below the limit of detection for this assay, and should be adjusted accordingly.

PLOS authors have the option to publish the peer review history of their article (what does this mean?). If published, this will include your full peer review and any attached files.

Reviewer #1: No

Figure Files:

Data Requirements:

Reproducibility:

References:

---

## [Editor Report · Decision Letter 1]

3 Dec 2021

Dear Prof. Murcia,

We are pleased to inform you that your manuscript 'Long-term adaptation following influenza A virus host shifts results in increased within-host viral fitness due to higher replication rates, broader dissemination within the respiratory epithelium and reduced tissue damage' has been provisionally accepted for publication in PLOS Pathogens.

Best regards,

Marco Vignuzzi, Ph.D.

Section Editor

PLOS Pathogens

Marco Vignuzzi

Section Editor

PLOS Pathogens

Kasturi Haldar

Editor-in-Chief

PLOS Pathogens

orcid.org/0000-0001-5065-158X

Michael Malim

Editor-in-Chief

PLOS Pathogens

orcid.org/0000-0002-7699-2064

Thank you for this revised version, all comments have been addressed.
---

## [Editor Report · Acceptance letter]

12 Dec 2021

Dear Prof. Murcia,

We are delighted to inform you that your manuscript, "Long-term adaptation following influenza A virus host shifts results in increased within-host viral fitness due to higher replication rates, broader dissemination within the respiratory epithelium and reduced tissue damage," has been formally accepted for publication in PLOS Pathogens.

Best regards,

Kasturi Haldar

Editor-in-Chief

PLOS Pathogens

orcid.org/0000-0001-5065-158X

Michael Malim

Editor-in-Chief

PLOS Pathogens

orcid.org/0000-0002-7699-2064